# Discrimination in In-Patient Geriatric Care: A Qualitative Study on the Experiences of Employees with a Turkish Migration Background

**DOI:** 10.3390/ijerph17072205

**Published:** 2020-03-25

**Authors:** Nazan Ulusoy, Anja Schablon

**Affiliations:** Competence Centre for Epidemiology and Health Services Research for Healthcare Professionals (CVcare), University Medical Centre Hamburg-Eppendorf (UKE), 20246 Hamburg, Germany; a.schablon@uke.de

**Keywords:** interpersonal discrimination, Turkish employees, migration background, care workers, in-patient geriatric care, nursing, residents

## Abstract

In most studies, nurses with a migrant background report experiences of interpersonal discrimination. These often occur in interaction with those in need of care. However, in Germany this topic has remained largely unexplored, although a large proportion of the employees in geriatric care have a migration background. The aim of the study was to investigate whether care workers with Turkish migration background in in-patient geriatric care are exposed to discrimination from residents. Furthermore, the reasons for discrimination, handling of discrimination and recommendations for in-patient geriatric facilities to avoid/reduce discrimination were examined. In a qualitative, explorative study, 24 employees with Turkish migrant background working in in-patient geriatric care were interviewed in 2017. The semi-structured interviews were evaluated using a qualitative content analysis according to Mayring. The majority (N = 20) experienced or observed discrimination. This occurred mainly in the form of xenophobic insults and rejections. They perceived visible traits (dark hair and eye color, clothing) as potential reasons. To deal with the discrimination, most of them temporarily left the scene. They recommend that institutions should primarily make the diversity of the workforce transparent to avoid/reduce discrimination. More research is needed about discrimination against care workers with migration background because discrimination may have serious psychological effects that impact employee retention and the quality of care.

## 1. Introduction

In light of demographic changes in Germany (owing in part to low birth rates and rising life expectancies), the number of people requiring care and the associated need for qualified specialists will increase significantly [1,2]. To be able to meet this demand, there are increasing efforts under way to recruit descendants of migrants already living in Germany for nursing professions. Efforts are also being made to recruit trained nurses from other countries of origin [3]. Nearly one in four professionals in geriatric nursing already has a migrant background [1]. The rise in the number of nursing staff with a migrant background is seen as positive as they are helping to close the nursing gap and thus are making a key contribution to the availability of nursing care. However, studies show that care workers with a migration background are subject to a wide range of discrimination in Germany [2]. Discrimination (Latin: discriminare) stands for "separating" or "making distinctions" and/or "segregating" [4]. Discrimination is an everyday phenomenon for carers with a migrant background, both in out-patient and in-patient care [5,6,7,8,9,10]. In this context, a distinction is made between institutional and interpersonal discrimination. In line with other publications, this paper takes up the interpersonal discrimination that emanates from those in need of care and is directed at caregivers. Interpersonal discrimination refers to “directly perceived discriminatory interactions between individuals—whether in their institutional roles or as public or private individuals” [11] (p. 301). Interpersonal discrimination includes prejudice (i.e., stereotypes, different assumptions about the abilities, motives and intentions of others on the basis of their race or ethnic origin) and discriminatory acts committed by individuals against persons from another race or ethnic group [11,12]. Discrimination on the basis of "race" or ethnic origin, along with gender, religion and/or belief, disability, age and sexual identity, is one of the six characteristics of discrimination listed in §1 of the General Equal Treatment Act [13]. 

Several studies suggest that discrimination can have a negative impact on physical and mental health and also wellbeing [14,15,16,17,18]. Numerous studies have indicated a relationship between the perception of discrimination and mental disorders such as depression and anxiety [15,19,20]. A meta-analysis revealed that the relationship between discrimination and mental health was more pronounced than the relationship between discrimination and physical health [21]. Studies also suggest that discriminatory experiences can have an impact on cardiovascular health [22] and suicide ideation/attempts [21]. The experience of discrimination, caused by unequal treatment, insults or threats, may be perceived as an assault on the victim’s own identity and result in diminished self-esteem [23,24]. Studies have also shown that the perception of discrimination can result in adoption of unhealthy lifestyles (e.g., poor diet, smoking, alcohol/narcotic/substance abuse) [18,21,25,26] or prevent healthy behaviors [17]. It can also create a hostile working environment [27]. Investigations have also revealed that it is likely that care workers exposed to discrimination are at greater risk of suffering from occupational diseases and injuries due to increased stress [28,29]. Behavior and attitudes towards work may also be adversely affected, resulting in a reduced work performance [30,31] and job satisfaction [32,33]. This in turn may correlate positively with the desire to leave the job or even the profession [32,34,35].

There are currently only a few empirical studies available in German-speaking countries that have examined the situation of nurses with a migrant background and their experiences of discrimination in everyday working life. Among the publications considered (mainly from the Anglo-American region or the United Kingdom), it is noticeable that the publications focus primarily on nurses who had trained in their home country and then migrated to a new country [7,9,36,37,38,39,40]. In contrast, publications involving nurses with a migrant background who have lived in Germany for decades or indeed since their birth and were not trained abroad are much rarer. This paper addresses this research gap. The aim of this study is to qualitatively record and present the interpersonal discrimination experienced by care workers with Turkish backgrounds employed in in-patient care. Specifically, the research question was whether employees with a Turkish migration background working in in-patient geriatric care have experienced discrimination by residents owing to the characteristic “ethnic origin”.

The study focused on the population group with a Turkish migration background, as they represent the largest group within the population with a migration background in Germany [41]. In addition, besides those from Poland and Russia, they form the largest group in the care sector [1]. Moreover, this population group is at a significantly higher risk of discrimination in many areas of life (access to work, housing and education) than other groups of origin (such as immigrants from Eastern Europe) [42]. 

## 2. Study Design and Method

To answer the research question, qualitative interviews were conducted from April to June 2017 in four in-patient geriatric care facilities (three in Hamburg and one in Bremen). The survey followed a semi-structured guideline, which covered four thematic areas: (1) form of experienced discrimination; (2) reasons for experienced discrimination; (3) dealing with experienced discrimination; (4) recommendations for care institutions to avoid or reduce discrimination. The following inclusion criteria applied to the interview participants: to be employed in residential care, to be themselves or one of their parents to be born abroad, not to have completed education (if any) abroad, and to be able to speak Turkish or German. Initially the plan was to consult only nurses with Turkish migration background. Owing to problems with recruitment and the fact that discrimination not only affects nursing staff but all personnel involved in providing care, employees in other jobs were also included. 

The care workers with Turkish migration background were not directly asked during the interviews whether they had personally experienced or observed discrimination as it was unknown whether they would perceive their own experiences as discrimination or not. As such, they were first asked to think about a time/situation where residents behaved differently towards them compared to their German colleagues. When they had such an experience, they were asked to give specific examples. This concerned discrimination that occurred in interaction with the residents and took the form of verbal or non-verbal violence (e.g., through insult, abuse, devaluation, ignoring or rejection). Physical violence was excluded. The interview then followed a more rigid structure. The following questions were asked: What do you think might have been the reasons for the unequal treatment? How did you deal with such situations? What could care institutions do to avoid or reduce unequal treatment?

The institutions were selected using the AOK Care Navigator (AOK-Pflegenavigator). All in-patient geriatric care facilities in Hamburg, Bremen, Berlin, Osnabrück and Lüneburg included in the list were contacted by e-mail and telephone and invited to participate in the study. Contact was established with participating employees with Turkish migration background with the aid of the institutions’ managers, who had informed their staff about the study. All participants gave their verbal consent prior to the interview and were informed that participation was voluntary and that they could terminate the interview at any time. With the agreement of the participants, the interviews were conducted at the relevant place of work, behind closed doors. It was possible to perform the interviews in German or Turkish, depending on preference, as the initial author conducted all interviews herself and also has a Turkish migration background. The interviews took between 5 and 31 min.

The interviews were recorded on audiotape and then transcribed. All available data were categorized according to the content of the four topics and evaluated with the help of a qualitative content analysis according to Mayring [43]. The analysis was performed with the software MAXQDA 11. 

All data in this trial was collected, analyzed and disclosed anonymously, following the terms of the German Federal Data Protection Act (BDSG) and the Data Protection Act of the City of Hamburg (HmbDSG). Therefore no ethical approval was required. The data is not publicly accessible. In line with the principles of data economy and for the protection of the anonymity of the participants, we did not obtain written consent for the interviews. Their verbal consent was documented in the interview protocol. The sensitivity of the subject demanded that no personal data that may identify the interviewee was collected.

## 3. Results

### 3.1. Study Population

A total of 24 interviews were conducted with employees in in-patient geriatric care (Table 1). Just under a third was employed as certified (healthcare) nurses and a quarter as geriatric nurse/healthcare nurses. All were female with the exception of one person. The average age of the participants was 37.3 (19–56 years). Around a third of them were younger than 35 (29.2%), while two were older than 50. The participants had been working in nursing for ten years on average. The duration of each participant’s employment in their respectively current institution varied between 0 and 20 years. Almost 60% of them were full-time employees (≥35 h). On average they had been living for over 27 years in Germany (10–48 years), and almost half of them were born in Germany. 

Of the 24 interviewees, most (N = 20) reported having experienced or observed at least one case of xenophobic discrimination from residents at their place of work that they believed was due to their ethnic origin. Three reported experiences of discrimination by residents that had nothing to do with their ethnic origin. They drew this conclusion on the basis of residents behaving in exactly the way towards German colleagues and that they were not viewed to be of Turkish origin by the residents. One person had neither experienced nor observed incidents of discrimination. The four participants were not included in the analysis.

The key statements made by the 20 participants, who reported having personally experienced or observed such incidents, are listed below. The categories most commonly named are described in detail. The other categories are only listed as defined examples in tables. The following topical fields were defined for the analysis:➢Forms of perceived/observed discrimination➢Reasons for discrimination➢Handling of discrimination (coping strategies)➢Recommendations for institutions for avoiding/reducing discrimination

### 3.2. Forms of Perceived/Observed Discrimination

The interviewees had been faced with a wide variety of forms of discrimination at their place of work (Table 2). The main forms stated were verbal attacks and insults of a xenophobic (“fucking foreigner”, “fucking Turk”, “Kanake” (a German word for people from German-speaking countries with roots in Turkey, Arab countries or Southern Europe), “darkie”, “headscarf woman”) and sexist (“Turkish bitch”, “cunt”) nature. An example quoted by one participant, who was training to become a certified geriatric nurse (Interviewee (IT) 1, aged 38) illustrates an extreme form of racial discrimination. She reported that she wanted to help a colleague who was involved in a dispute with a resident, who was then reported to have insulted the trainee by saying “the likes of you were not allowed to walk around here in the old days”. The director of the institution then had the resident relocated to another facility on the same day and terminated the nursing care contract. Participants reported rejections and refusal from residents occurring almost as frequently. Care workers were told with humiliating language such as “No, I don’t want you. Go away, you’re a foreigner”) that resident rejected receiving care from a “foreigner”. Participants also described rejections expressed in non-verbal form. One nursing assistant (IT 14, aged 41) reported one resident not talking to her for two years. She was ignored by this resident for two years. A deputy senior certified geriatric nurse (IT 6, aged 28) reported noticing that a Greek resident would not let herself be touched by the participant. She believed that the resident was expressing her attitude towards her through her body language. She perceived this behavior to be highly stressful; this also made providing nursing care more difficult. 

### 3.3. Reasons for Discrimination

The interviewed employees considered the reasons for discrimination to be diverse (Table 3). The most commonly named causal factors were externally visible traits (dark eyes, dark hair) and religious traits (headscarf) (N = 10). It was also noted, however, that residents reacted to Germans with dark eye and/or hair color in the same way as to staff with Turkish migration background. Reported experiences varied in relation to the headscarf. A certified geriatric nurse (IT 11, aged 41) said that the residents associated the headscarf with traditional headwear of nuns and that she was referred as the “holy sister” by them. Others in turn reported very negative experiences. Two female participants reported an experience during their training (in out-patient care) when the client denied them access to the apartment. A relevant quote: “You’re not coming in here with your scarf. Either you take it off and come in or you can go home” (IT 20: Certified nurse, aged 45). The situation had a major influence on their professional career. She said she would never like to work in out-patient nursing where nobody could intervene to protect her. This is an example from out-patient care, but one that is so significant that it must not be left unmentioned. Another factor that was frequently mentioned was the generational aspect. In this context, the lack of social contact was referenced between the elder generation of Germans and people of Turkish or other foreign descent. Reports also mentioned that there were no problems with those who had previously had contact with people of Turkish migration background (e.g., at the workplace). They believed it was due to the prejudice-based upbringing of the elderly. They were always told by their parents that foreign children were dirty and that they were not to play with them. They also attributed it to experience during the War. There were reportedly residents who had lived through the War and still had an affinity to Nazi ideology. One participant came to this conclusion based on a statement from one of the residents: “I’m German. I’m exceptional” (IT 14: nursing assistant aged 41).

### 3.4. Handling of Discrimination (Coping Strategies)

Methods of handling discrimination also varied greatly (Table 4). Most of the respondents reported leaving the room in conflict situations and leaving the resident alone for a while. After a short while, once the resident had calmed down, they went back in. Others preferred to hand the problematic resident temporarily over to a colleague. If there was no colleague to hand nursing duties over to, nursing activities were sometimes suspended. Others still set clear boundaries and refused to provide care to the resident in question. One trainee certified nurse (IT 8, aged 19) justified this by pointing out that residents who rejected one of the staff resorted to shouting. The ‘shouting’ was then reported to cause unrest among the other residents. She was also worried about a repeat of the situation. She also did not want to unnecessarily torment or agitate the resident. When questioned about their specific responses to discrimination, many participants reported that they ignored discriminatory statements and tried to maintain a professional attitude. Racial or xenophobic comments like “fucking foreigner” or “fucking Turk” were ignored or simply laughed away. Many also rationalized it and attributed the statements and actions of the residents to their diseases (e.g., dementia). They considered it important to not forget that the residents were sick and therefore also to not take offence. They believed that the residents were not even aware of what they were saying. They would forget about it after just a few minutes and wanted a kiss.

### 3.5. Recommendations for Institutions for Avoiding/Reducing Discrimination

Regarding the possible measures that care institutions can take to avoid or reduce discrimination, the respondents recommended three courses of action (Table 5). Firstly, transparency regarding the ‘multicultural’ nature of the workforce was recommended. The ethnic diversity of the workforce should be mentioned on the institution’s website and information materials provided. Transparency regarding the workforce structure would help to ensure that people who do not wish to be cared for by staff with differing ethnic backgrounds do not choose this institution. Secondly, the manager should give special attention to the situation when disputes arise between care workers and residents (due to ethnic origin) and take into account the preference of the resident (care by a native nurse) should be taken into consideration, for example when organizing duty rosters. This would prevent both the nurse and the resident from being exposed to undue stress or overwhelming situations. Thirdly, the need to ensure that adequate time is dedicated to allowing the nursing staff and residents to get to know one another was also specified (e.g., at lunch instead of intimate care). This would allow uncertainties regarding interactions with one another to be eliminated and trust to develop.

## 4. Discussion

The results of this study suggest that care workers with a Turkish migration background are exposed to discrimination by residents in in-patient geriatric care. While most of them spoke of personal experience, some reported witnessing discrimination. The most common scenario named was verbal attacks and insults of a racist or xenophobic nature. These findings are consistent with the findings of other studies where nurses who were direct or later-generation migrants were frequently exposed to disparaging comments from residents [7,44]. Statements like “the likes of you were not allowed to walk around here in the old days” or “go back where you came from” were rare. Such statements can lead to hostile sentiments among those affected, however [45]. Respondents also reported events where the residents refused to receive care from them both verbally (“I don’t want you” and non-verbally (signaling an unwillingness to be touched). Direct rejection by care home residents and/or patients of nurses with a migration background was also shown in other studies [32,35,37,46,47]. 

In this study, care workers with Turkish migration background clearly identified their ethnic origin or ethnic markers as the main reason for the discriminatory statements and actions of the residents. Over half said that their experiences were related to their visible markers; these include traits such as skin color, eye color or clothing style (headscarf) [48]. This is consistent with the results of previous studies, where residents in need of care also expressed racist sentiments towards care workers of visible ethnicity and refused to accept care from these workers [49,50,51,52,53]. In the study by Wheeler et al. [7], rejection by patients was deemed to be the strongest discriminatory behavior. According to Salentin [54], visible traits of foreignness amplify discriminatory behavior. Where people believe that they have experienced discrimination due to visible markers, this may adversely impact their physical and psychological health [55,56]. Some of the respondents attributed the racist comments and actions of the residents to age and the generational factor. The European status description regarding intolerance, prejudice and discrimination showed that over-65s were significantly more inclined towards xenophobia, Islamophobia and racism compared to younger generations [57]. Rather than taking discrimination personally, participants looked on these incidents within the historical and social background of the residents. As with other studies, respondents attributed it to factors such as education and the era in which the residents had grown up [58,59]. Discrimination and prejudice expressed by (predominantly) older residents was also attributed to the lack of contact [60] and lack of experience with other nationalities [51]. Geriatric care homes are a place where many elderly native Germans encounter people from other countries for the first time. In their study on xenophobia, Geißler et al. [61] considered both the contact hypothesis [62] and the conflict hypothesis [63] to be confirmed; in other words, a lack of contact with people of foreign origin encourages resentment. It is probably true that older people have less contact with migrants but have had longer exposure to stereotyping messages and are more likely to harbor racist attitudes. Stereotypes and prejudice against a group may affect behavior towards that group [64]. As also stated in the qualitative study by Stagge [60], this rejection was attributed to the wartime experiences of the residents and an aversion towards foreigners acquired as a result of these. According to Kistner, the process of socialization during the National Socialist dictatorship plays a defining role. In their study, Wilhelm & Zank [65] discovered that the wartime experiences of care patients (war trauma) had an adverse impact on the daily routine of nurses, for example defensive attitudes towards nursing in general, especially towards male or foreign nurses. 

In handling with discrimination, there were both passive and active management and coping strategies. As in the study by Salavati et al. [66], many of the respondents left the room and returned after a short time. The ‘leave and return’ strategy was also recommended by Harwood [67]. Others requested assistance from a colleague. In situations that seem dangerous, one of the 12 basic rules of de-escalation is, for instance, to bring in a colleague to provide support and ensure personal safety [68]. Participants also specified responses such as “failure to provide care”, which is perhaps indicative of a possible care deficit. According to Wheeler et al. [7], discrimination may create a hostile environment and have an adverse impact on patient care and safety [69,70]. The forms chosen by some represent an active coping strategy, which can help to mitigate or reduce the negative impact of perceived discrimination on psychological health [71]. Others in turn adopted a passive strategy whereby they ignored or rationalized the discriminatory statements and actions, as described in the study by Berdes and Eckert [72]. It was often attributed to the mental confusion or illness (e.g., dementia) of the residents and not interpreted as discrimination. According to Ivanova [73], passive action constitutes an attempt to disassociate oneself from the unpleasant aspects of discrimination without attempting to combat discrimination. A passive management strategy is commonly applied if it is assumed that circumstances will not change [74], such as the residents’ conditions. 

In terms of what institutions can do to avoid or reduce discrimination, respondents provided helpful recommendations. They believed that greater transparency regarding the ‘multicultural’ nature of the workforce may be helpful. For example, the institution’s website and information materials should explain how diverse the workforce is. This allows potential residents and relatives to be informed about whether the home is the right place for them or their relatives before they even enter the care institution. While this proposal sounds very useful to begin with, given that migrants already account for a significant share of all nurses in Germany, it is possible that the institution may harm itself with this course of action. Prejudiced customers would likely reject the care facility. Another recommendation was such as responding to the residents’ objections regarding nurses of a certain ethnic origin. However, in the study by Mapedzahama et al. [37], it was shown that managers replacing nurses with migration background rejected by patients with native nurses was not viewed as supportive, but rather evasive. Respecting these demands was viewed as encouraging the racist behavior of the patient [37]. This behavior may be viewed by care workers with a migration background as institutional support for racism and discriminatory behavior, even if it was not so intended [75]. Dealing with racial discrimination is not an easy task for employers. Firstly, they are required to protect their employees against racist attacks, which the German Social Code VII defines as an “occupational accident”. Secondly, expressing tolerance of such behavior may be viewed among staff with migrant background as institutional support for racial discrimination [75]. The matter should therefore always be discussed with the care worker beforehand. There was also a recommendation of dedicating more time to familiarization when residents move in. Studies have shown that migrants working in the nursing sector are initially not well-received by older patients, but that the reluctance usually disappeared when they got to know one another better [53,76]. Through frequent contact with members of other groups (care workers with a migration background), it may be possible to reduce prejudice and stereotypes against this group and to develop trust, as mentioned above.

## 5. Strengths, Limitations of Study and Recommendations for Future Research

The greatest strength of this study is that it enriches the literature on discrimination experienced by migrants working in the nursing sector as this topic had only been addressed inadequately in Germany. The findings may also be useful for many decision makers in the political realm and for actual workers in care institutions. It also enabled inclusion of those with a poor command of the German language, who are heavily under-represented in studies. As the initial author was herself of Turkish descent, it was possible to conduct the interviews in a language preferred by the participants. 

However, this study was subject to a range of limitations. The study was able to illustrate the experiences of employees in in-patient geriatric care with a Turkish migration background, as this group is strongly underrepresented in the literature. But in view of the small size of the sample, which consisted of only four in-patient geriatric care facilities, the results do not permit any generalization. Further research, in the form of a quantitative survey, with more participants and care institutions is necessary to support the results. Furthermore, the identity of the interviewer, who is herself of Turkish descent and is a native Turkish speaker, may have influenced the responses from the interviewees. It is noted that the interaction between the researcher and interviewees may have advantageous or adverse effects on the process of data collection [77]. The same ethnic origin can create trust and contribute to the creation of an open and trusting atmosphere in conversation [78,79]. On the other hand, it can be seen that interviewees who have the same ethnic background tell different things to those who do not have the same ethnic background.

The study was limited to the experiences of care workers of Turkish descent. A future study may involve nursing personnel of different ethnic origins. A study of the views of different groups such as residents and nursing personnel with and without history of migration in their family would help to identify the various facets of the problem and could be helpful in developing intervention strategies at several levels. 

## 6. Conclusions

The results of this study show that care workers of Turkish descent in nursing frequently experience discrimination at their place of work. As a result of demographic change, the importance of care workers with a migration background in Germany will increase. If no action is taken to avoid or reduce discrimination, the quality of care may suffer and migrant care staff may abandon the nursing profession. More research is needed about discrimination against care workers with a migration background.

## Figures and Tables

**Table 1 ijerph-17-02205-t001:** Characteristics of the sample (N = 24).

Characteristics	Categories	N (%)
**Age**	<35	7 (29.2)
	35–50	15 (62.5)
	>50	2 (8.3)
**Gender**	Female	23 (95.8)
	Male	1 (4.2)
**Marital status**	Unmarried	5 (20.8)
	Married	11 (45.8)
	Divorced	8 (33.4)
**Duration of residence in Germany (years)**	10–20	6 (25.0)
	21–30	11 (45.8)
	>30	7 (29.2)
**Born in Germany**	Yes	11 (45.8)
	No	13 (54.2)
**Professional status**	Certified (healthcare) nurse in senior position	4 (16.7)
	Certified (healthcare) nurse, not in senior position	3 (12.5)
	Geriatric nurse/healthcare nurse	6 (25.0)
	Nurse without formal qualification	3 (12.5)
	Trainee	2 (8.3)
	Other (catering service, ward service/kitchen assistant, gap year volunteers, etc.)	6 (25.0)
**Terms of employment** (Missing values 4)	≥35 hours a week	14 (58.3)
	<35 hours a week	6 (25.0)

**Table 2 ijerph-17-02205-t002:** Forms of perceived/observed discrimination.

Category	Defined Example
Racist verbal attacks/insults	“’Fucking foreigners!’ They use that word a lot.” (IT 3: Medical nursing assistant, aged 48) “One of the residents always says ‘fucking foreigners’ or ‘you darkie’ and ‘I’m going to punch you’. In my view these words or sentences are racist.” *(IT 2: geriatric nursing assistant, aged 36)* “Because I wear a headscarf, she [the resident] always called me the ‘headscarf woman’ behind my back.” *(IT 5: Service assistant, aged 36)*
Sexist verbal attacks/insults	“They [the residents] said things like ‘Turkish bitch’.” *(IT 14: Nursing assistant, aged 41)* “She [the resident] called her [employee wearing a headscarf] things like ‘cunt’.” *(IT 9: Gap year volunteer, aged 20)*
Rejection	“One [of the residents] said for example, ‘She [a Turkish colleague] is a foreigner, I don’t want her.” *(IT 17: Medical nursing assistant, aged 39)* “She [the resident] screamed ‘Get out, I don’t want you. Go away (...), you’re a foreigner’.” *(IT 10: Nursing assistant, aged 42)* “She said, ‘oh, the one with the headscarf. I don’t want to be cared for by you’.” *(IT 8: Trainee nurse, aged 19)*
Hindering care	“There’s a Greek resident who always says ‘Turks no, Turks, no’. (...) There are always problems with her when tending to her needs and transferring her. You can just see that she doesn’t want to touch someone.” *(IT 6: Deputy senior certified geriatric nurse, aged 28)*
Harassment	“With one of the residents I actually always had this problem where she would always do or want the opposite because I wear a headscarf. If I gave her a bit of bread, she always wanted a different one.” *(IT 5: Service assistant, aged 36)*
Insinuating incompetence	“Behind ones back you hear them say negative things about my colleague who wears a headscarf: ‘The one with the headscarf, she’s useless.’” *(IT 13: Certified geriatric nurse, aged 29)*
Accusations	“Then she claimed that I supposedly wasn’t Turkish, but Syrian, and that I was a liar. She claimed that I was a refugee.” *(IT 5: Service assistant, aged 36)*

IT = Interviewee.

**Table 3 ijerph-17-02205-t003:** Reasons for discrimination.

Category	Defined Example
Ethnic and religious traits	“We only have dark eyes and dark hair, but still they always say ‘fucking foreigner’.” *(IT 3: Medical nursing assistant, aged 48)* “Maybe how we look. Because we don’t have blond hair and blue eyes.” *(IT 24: Certified geriatric nurse, aged 26)* “We also have Germans, blond-haired ones. He [the resident] hadn’t said the words ‘fucking foreigner, darkie’ to them yet, but he had to us.” *(IT 2: Geriatric nursing assistant, aged 36)* “He [the resident] doesn’t like me anyway. He doesn’t like dark-haired people, and I have dark hair.” *(IT 4: Health and nursing assistant, aged 49)* “One of the residents didn’t want me (...). She didn’t want a girl who wore a headscarf.” *(IT 22: Certified geriatric nurse, aged 25)* He [the resident] is obsessed with dark hair. For him, they’re all foreigners. You might have our dark-haired boss walk in and he’d still scream ‘fucking foreigner, get out!’” *(IT 3: Medical nursing assistant, aged 48)*
Generational aspects	***(Lack of) contact*** “These days it’s the postwar generation that never really had anything to do with the foreigners. You had the Turks here and the Germans there. But I knew three residents who had worked with Turks and they said: ‘They [Turkish colleagues] were always nice, always friendly’.” *(IT 6: Deputy senior certified geriatric nurse, aged 28)* “The older generation of Germans, they didn’t have this close contact with one another.” *(IT 20: Certified nurse, aged 45)*
	***Prejudice/prejudice-based upbringing*** “The older [residents] had their prejudices, it wasn’t particularly nice.” *(IT 6: Deputy senior certified geriatric nurse, aged 28)* “They [the residents] would say things like ‘I don’t want her, she’s dirty.’” *(IT 14: Nursing assistant, aged 41)* As children they’d be told, ‘Foreigners are shit, disgusting, don’t be friends with them.’” *(IT 3: Medical nursing assistant, aged 48)*
	***War experiences*** “We’ve got one [resident] at the moment who always says ‘I’m German, I’m exceptional’ (...) And so I say ‘good for you that you’re German. I’m almost German too. I was born here. [The resident’s answer to this.] ‘But I’m a proper German. I look like one’.“ *(IT 14: Nursing assistant, aged 41)* “She [resident] talks a lot about Hitler. You can tell she used to be a supporter. (...) She’s stuck in time.” *(IT 12: Domestic services assistant, aged 38)* “Some have also been shaped by the war. While they don’t talk about it, some were in favour of the situation back then [National Socialism]. They still have the same attitudes from back then.” *(IT 24: Certified geriatric nurse, aged 26)* “We have two residents here who say ‘I hope you get to experience war.’” *(IT 12: Domestic services assistant, aged 38)*
Language/ communication barriers	“She doesn’t understand me and I don’t understand her.” *(IT 10: Nursing assistant, aged 42)* “Some of the patients [residents] who were mentally still fit recognize from my accent/dialect that I’m not German. They say things like ‘You need to speak clearly’.” *(IT 21: Nursing assistant, aged 47)*
Need for help	“Now she’s not quite as mobile and needs more help. I think that is a big reason why she is being a lot friendlier now.” *(IT:* Trainee nurse, aged 19)
Physical and psychological well-being	“I don’t take it so seriously. (...) After all, he [the resident] is sick.” *(IT 2: Geriatric nursing assistant, aged 36)* “There are sick people here. They have different diseases. You mustn’t forget that. Maybe they don’t really mean what they’re saying.” *(IT 3: Medical nursing assistant, aged 48)*

IT = Interviewee.

**Table 4 ijerph-17-02205-t004:** Handling of discrimination (management strategies).

Category	Defined Example
Leaving the room	“At that moment, I leave the resident in peace and come back ten minutes later. By that time, they [the resident] will have calmed down.” *(IT 3: Medical nursing assistant, aged 48)*
Removing oneself from the situation	“It becomes really exhausting if the resident is verbally attacking you in Greek the entire time and making it difficult to provide care. (…) So I then kept away from her. (...) If she doesn’t want me to come then I won’t go to her. (…) This defensive attitude and the feeling of working against the current is a real burden.” *(IT 6: Deputy senior certified geriatric nurse, aged 28)*
Involving a colleague	“When it got really extreme and you couldn’t even get to the resident, I left the room (...), went to a colleague and asked him if he could go and take care of her.” *(IT 6: Deputy senior certified geriatric nurse, aged 28)*
Withdrawing/abandoning	“In that moment [when she is being verbally attacked], I just leave them [the resident] in peace and leave. Either they’re taken care of when someone is there, someone else, and if not, well, that’s the way it is.” *(IT 2: Geriatric nursing assistant, aged 36)*
Ignoring	“I’ve just always ignored comments like ‘oh, that fucking Turk’. *(IT 8: Trainee certified nurse, aged 19)* “What else can I do, I laugh myself stupid and then get back on with my work. I simply ignore them [the resident] and then it’s all good.” *(IT 3: Medical nursing assistant, aged 48)*
Rationalization	“You can’t be angry at them [the residents]. These are sick people, they need our help.” *(IT 4: Nursing assistant, aged 49)* “I then think about what she [the resident] must have experienced in the past or what kind of time she lived in and why she thinks that way.” *(IT 14: Nursing assistant, aged 41)* “They [residents with dementia] say one thing one moment, then they’ve forgotten this the next. They don’t really mean it. (…) Two minutes later they want to give us a kiss.“ *(IT 7: Ward assistant, aged 56)*
Confrontation	“There’s nobody but this ‘fucking foreigner’ to wipe your backside.” *(IT 3: Medical nursing assistant, aged 48)* “One of the patients [residents] said, ‘you need to learn German’. But he was laughing while he said it. So I retorted that he needs to learn Turkish. I’ve been coming here for years.” *(IT 21: Nursing assistant, aged 47)*

IT = Interviewee.

**Table 5 ijerph-17-02205-t005:** Recommendations for institutions for avoiding/reducing discrimination.

Category	Defined Example
Transparency	“I think it’s important to clarify everything beforehand when relatives become involved or new residents move in. (…) Maybe advertise more that we are multicultural. That the nursing services in the residential areas aren’t just provided by Germans, but from lots of people from different countries. *(IT 24: Certified nurse, aged 26)* [More transparency] “The children know how their mother is. One of the daughters always says ‘Mum, a person is a person, whether they’re white, black, Turkish or German.” *(IT 6: Deputy senior certified geriatric nurse, aged 28)*
Staff assignment	“Maybe they shouldn’t send any more dark-haired nurses to care for this resident. There’s no need to unduly stress or overwhelm the resident. This is how we should avoid it.” *(IT 4: Nursing assistant, aged 49)* “One solution would be to perhaps stop working with residents where there is an antipathy and provide for another resident.” *(IT 20: Certified nurse, aged 45)*
Relationship development	“Near the start, one of the residents said to me, ‘Oh, the one with the headscarf. I don’t want to be cared for by you.’. (…) It’s not like that at all now. We get along much better now.” *(IT 8: Trainee certified nurse, aged 19)* “We used to have a resident who always used to say ‘oh shit, fucking foreigner’s back again’. (…) I told my colleagues that no-one else except me should care for this lady and that I wanted to look after her alone. (…) Later I was her best friend.” *(IT 3: Medical nursing assistant, aged 48)*

IT = Interviewee.

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
