# Peer review of "Discrimination in In-Patient Geriatric Care: A Qualitative Study on the Experiences of Employees with a Turkish Migration Background"

_ijerph, 2020, doi:10.3390/ijerph17072205_

Round 1

Reviewer 1 Report

In this article the authors cover a very important topic and fill a knowledge gap in the literature. The research question is highly relevant and topical, the methodology is stringent and well explained and the presentation of the results is clear and easy to understand. Only the discussion part needs some more revision, since it is to some extent redundant with the results section and could be more concise.

Still, one general issue I have with the paper concerns the concept of “racial discrimination”. I am not convinced that what the authors convincingly describe as discriminating behavior towards care workers with a migration background is indeed correctly categorized as “racism”. I would suggest to reflect, if the topic could not be more precisely called “xenophobic discrimination” (with xenophobia in the sense of “hatred of foreigners”, the German “Ausländerfeindlichkeit”). After reading the article I have the impression that most instances of discrimination mentioned in the article are not based on the notion of a construction of otherness based on race, but rather a notion of otherness based on being “foreigners”. This is particularly apparent on p. 12, where the story of a German nurse is told, who is discriminated in the UK. Here very clearly this is not racism but xenophobia. Most of the quotations presented in the text and the tables similarly express xenophobia, not racism: The trigger for the construction of otherness is the assumption, that the caretakers are “foreigners”. I would therefore encourage either to render the topic of the paper more precisely as “xenophobic discrimination”, or more broadly as “discrimination” and then specify in the particular examples what is the motive for the discrimination.

Otherwise, I have only very minor comments:

The use of the verb “handle” seems not totally correct to me. Since there are also some other small grammatical uncertainties, I would like to recommend having a native speaker language-edit the article.

Introduction: I would recommend to at least briefly also mention the debate concerning violence against the recipients of geriatric care.

p.2, 13: The phrase “people of different ethnic origins“ needs some clarification: different from what or whom? I understand that it is meant to say “different from care workers without a supposed migration background”, but it could also be understood to meant “a number of different ethnic origins”. Please clarify.

p.2, 35-36: The sentence on the efforts to attract more care workers with a migration background would benefit from being supported by a reference.

p.2, 39: The use of the word “will” irritates a bit here; it makes it look as if this was only a study protocol and the actual study has not been conducted yet. I would suggest writing “Therefore, we approached this topic in an explorative, qualitative study.”

p.2, 41-42: Grammar: remove the “with”: “Do care workers in inpatient geriatric care with Turkish migration backgrounds experience racial discrimination from residents?”

  1. 3, 67: This comment concerns the discussion about the underlying meaning of the German word “Rasse”: The German word “Rasse” has a distinctly different meaning than the word “race”. The German “Rasse” implies a biological notion and is (different from the English “race”) not a category of socio-cultural self-ascription. Therefore the use of the concept “Rasse” has for long been criticized and is largely avoided in the social sciences (but not in the law, which uses the term somewhat naively and has been criticized for that). See for instance: AG gegen Rassismus in den Lebenswissenschaften, 2009.

In your reference to the law you put the word “race” in quotation marks, which I applaud. I would suggest explaining to the non-German readers why, maybe in a footnote.

p.3, 70: Here I understand that you use the English concept “race”, and not “Rasse” (hence no quotation marks). This might not become clear to everybody. Also, very often people’s self-ascription and the labels they receive from others disagree.

I would therefore recommend to slightly change this sentence to: “[…] people are treated less favourably than others due to their supposed race or their supposed membership of a different population group.”

p.3, 83: Could you please give more details on the selection criteria? How were the institutions selected from the navigator: was it randomly, did you have certain criteria for the selection etc.?

p.4, Table 1: I assume that the duration of residence is given in years. Please add this information. Also, I would recommend to include a separate line for “born in Germany (yes/no)” and only elaborate on the duration of residence for those not born here.

p.4, 108: “the experiences” is misleading here. I understand you to say that in addition to the 20 respondents who reported racial discrimination, three more reported attacks from residents unrelated to racial discrimination. I would therefore change the sentence to: “Three reported experiences of violence and abuse by residents that had nothing to do with their ethnic origin.”

p.12: Table 2 names a third approach to avoid racial discrimination of employees, “relationship development”. This is not mentioned in the text below the heading 3.5. Could you give more information on that in the text? It seems interesting and I would like to know more about it.

p.11, 215: I would not use “&” in the text.

p.11, 211-228: These paragraphs summarize the results. I find this a bit redundant and would recommend to focus more on the contextualization of your findings within the literature.

p.13, 311: I have the impression that the authors are over-critical with their work. After reading the article I see no reasons to doubt the “credibility” of the finding. Further research and bigger numbers might lead to better generalizability; but credibility is not an issue here.

p.13, 312: Also, I would disagree that the qualitative approach only “illustrates” the experiences of the respondents. “Illustrating” a problem is what e. g. a journalist might do in a reportage. The authors, in contrast to that, used a stringent method of data collection and analysis based on clearly outlined theoretical assumptions. As a result, what the article does is a presentation of valid scientific findings, which are illustrated by selected quotations but are built on material much thicker and deeper. Therefore, I would like to encourage the authors not to be overmodest.

p.13, 315-317: I applaud the authors for considering the interviewer’s person as an influence. While the positive influences are obvious as outlined in the paper, I would want to recommend some more elaboration on the negative influences that you suspect. What do you think, how might the interviewer’s own migration background impair the data she collected?

p.13, 329: The last words of the sentence seem to be out of place. I would delete “because discrimination”.

Author Response

Thanks for the advice.

Reviewer 2 Report

This qualitative survey with n of 20 addresses an important topic, perceived discrimination in immigrant health  care providers in Turkey.

here are my comments:

1- the author need to explain how low n impacts the results of a qualitative study like  this.

2- discrimination in this study is not all due to race. it is also due to nativity. so, the term racial should change to discrimination due to race, ancestry, and nativity 

3- there is a huge literature that is not included. Please see the work by Gee and Williams on racism and discrimination. 

4- longitudinal studies have shown that racial discirmination predicts future symptoms of anxiety and depression. also, it predicts suicidal ideation, obesity, heart disease, social isolation, etc. these to be cited and discussed.

Author Response

Thanks for the advice.

Reviewer 3 Report

The first sentence of the abstract seems overly strong without support – “Violence in nursing is no longer an unknown phenomenon.”  First, it is unclear what it means “violence in nursing” – do the authors mean “violence by nursing staff”?  If so, this statement creates stereotypes toward nurses.  Also, why is it unknown?  What made it known?  This statement should be carefully rewritten and if it’s kept, it should be put in the main text follow by detailed supporting evidence of the statement. 

The second statement in the abstract is even worse – it linked violence by nursing staff (if that’s what the authors meant) to someone with migration background – “However, in Germany this topic remained largely unexplored, although a large 13 proportion of the employees in geriatric care have a migration background.”  Also, it’s unclear about the logic of the statement: Why is it that “although” employees have a migration background, Germany has not explored this problem?

Moving on to the third sentence, the authors mentioned “racial discrimination” all of a sudden – From the first two sentences I thought the authors were looking at violence performed by nursing staff?  All these mean that the authors have major problem expressing and connecting their ideas well.

The introduction reads a little better, clarifying upfront that they are looking at “violence AGAINST employees in nursing professions”.  However, violence and racial discrimination should be discussed separately – racial discrimination is not necessarily a type of violence.  Rarely does researchers say discrimination is violence – it could be, but the two concepts are different.  The authors need to clarify why they mention violence and racial discrimination side by side in their study context.   The authors also did not provide citation why racial discrimination is a form of violence (line 6). 

Try to avoid statement like “For Germany, there are hardly any reliable statistics and surveys on (racial) discrimination against care workers with a migration background.” Have you performed a thorough search and found no/little studies in Germany in this topic? If so, what database have you searched and how hard have you tried to search for these articles? Simply saying that “there are hardly any” is not a scientific way of arguing the significance of the current research.  

I did a quick search (1 minute) and I already found a few articles that the author should at least discuss:

Theobald, H. (2017). Care workers with migration backgrounds in formal care services in Germany: a multi-level intersectional analysis. International Journal of Care and Caring1(2), 209-226.

Inowlocki, L., & Lutz, H. (2000). Hard labour: The ‘biographical work’of a Turkish migrant woman in Germany. European Journal of Women's Studies7(3), 301-319.

Even if it’s true that there is no research in Germany on racial discrimination in inpatient geriatric care workers, there are plenty related topic, such as one on migrant domestic workers

Aulenbacher, B., Innreiter‐Moser, C., & Palenga‐Möllenbeck, E. (2013). New maids–new butlers? Polish domestic workers in Germany and commodification of social reproductive work. Equality, Diversity and Inclusion: An International Journal.

Something about migrant workers working in health care in general in Europe

Walsh, K., & O'Shea, E. (2009). The role of migrant care workers in ageing societies: Context and experiences in Ireland.

Or just general theories on discrimination.  Given that this is a qualitative study, the authors should have thorough theories support for their inquiry. Simply going out and interviewing people only warrant a news report, rather than a scientific research article.

In the research questions, try to increase clarify by using the right terms. For example “What do they believe is the trigger for this?” --- what is “this?  How do they handle such situations? – what are the “situations”.  Operationally define all the terms, rather than assuming people know what everything meant.

In section 2, please delete “To date, no research project in Germany has systematically studied experiences of racial discrimination of inpatient geriatric care workers with Turkish migration background; an exploratory, qualitative study design was therefore deemed to be suitable.” This is not a good justification of the method at all.

The manuscript needs some restructuring.  In section 2, literature review of previous studies should go to section 1.  In section 3, study population should go to section 2, rather than results.

The authors said they used content analysis, but did not detailed their analytic procedure. Please clarify how transcripts were coded and how themes were identified. Provide citation to support their analytic method whenever possible.  

“The results showed that personal experience and observation of racial discrimination were not related to age.”  - You are not doing quantitative analyses and this is not part of your research questions; therefore, it should not be stated this way.

In the results under “ethnic and religious traits”, I found that the examples are mostly about appearance and have nothing to do with religion.  Please correct the category label to reflect the category correctly.

Author Response

Thanks for the advice.

Round 2

Reviewer 3 Report

I appreciate the authors' work in revising the manuscript, particularly the introduction. The introduction now reads much better and much more logical than the previous version. The method is described with much more clarity and the discussion is aligned with the results.